# Ring-Core Fluxgate Sensor for High Operation Temperatures up to 220 °C

**DOI:** 10.3390/mi13122158

**Published:** 2022-12-07

**Authors:** Kaixin Yuan, Aimin Du, Lin Zhao, Shuquan Sun, Xiao Feng, Chenhao Zhang, Yiming Zhang, Huafeng Qin

**Affiliations:** 1CAS Engineering Laboratory for Deep Resources Equipment and Technology, Institute of Geology and Geophysics, Chinese Academy of Sciences, Beijing 100029, China; 2Innovation Academy for Earth Science, CAS, Beijing 100029, China; 3College of Earth and Planetary Sciences, University of Chinese Academy of Sciences, Beijing 100049, China; 4Faculty of Information Technology, Beijing University of Technology, Beijing 100124, China

**Keywords:** fluxgates, high temperature, drilling device, magnetic sensing, ring-core geometry

## Abstract

Fluxgate sensors are key devices for magnetic field surveys in geophysics. In areas such as deep drilling, fluxgate sensors may have to operate steadily at high temperatures for a prolonged period of time. We present an accordant ring-core type fluxgate sensor that is stable up to 220 °C. The high temperature consistency is achieved by using an Fe-based nanocrystalline magnetic core, PEEK structural components, an epoxy resin wrapping, as well as a broadband short-circuited working mode. The sensor was characterized at various temperatures up to 220 °C by evaluating impedance, hysteresis, permeability and sensitivity. We found a sensitivity of approximately 24 kV/T at 25 °C with an acceptable temperature coefficient of 742 ppm/°C throughout the range. The variation law of magnetic characteristics and their influence mechanism on output amplitude and phase are discussed.

## 1. Introduction

Fluxgate sensors are widely used in geophysical survey and space exploration, as well as electrical engineering and navigation [1,2,3,4,5]. Among all methods of measuring quasi-DC vector magnetic fields, they are well-known for their small size, low power consumption, low noise level and high robustness [6,7,8]. Fluxgate sensors measure a magnetic field through the fluxgate effect of the magnetic core. If the permeability of the core material is changed, the flux changes, and voltage is induced in the induction coil, deflecting the magnetic field [7]. Depending on the shape of the core, fluxgates can be roughly divided into several categories, including rod-shaped-core fluxgates, Vacquier-type fluxgates using a dual rod-shaped core, Goubau-type fluxgates using a ring-shaped core [9], race-track-core fluxgates, etc.

The most commonly used core materials of fluxgates are soft magnetic alloys. These alloys range from iron and low carbon steel through silicon-iron alloys to high nickel-iron and cobalt-iron alloys. The permeability of most soft magnetic alloys varies with the strength of the magnetic field applied to it [10]. For fluxgate applications, the ideal material can only carry an upper rated magnetic flux, beyond which it saturates, similar to a gate. When in saturation, the permeability decreases sharply compared to the large initial permeability. By measuring the change in permeability by various means, the value of the external vector magnetic field can be determined [7].

In special scenarios, such as deep drilling and in situ solar magnetic field surveys, the ambient temperature can be relatively high [11,12]. Therefore, the components, especially the magnetic cores, require high temperature reliability. However, most of the classic core materials [13,14] will fail at high temperatures over 200 °C due to drastic changes in magnetic properties, caused by exceeding the Curie temperature, or by irreversible destruction of the crystal structure. Some models of Fe-based nanocrystalline materials have a Curie temperature of nearly 600 °C. Their crystal stability is also better than that of amorphous alloys, and the crystallization temperature exceeds 500 °C. More importantly, their good soft magnetic properties make them suitable core materials for fluxgates [15]. Several experimental fluxgate magnetometers can withstand high temperatures up to 180 °C and 250 °C [16,17]. Both of the approaches are based on Vacquier-type fluxgate sensors, using the same core material of Ferro-based nanocrystalline Vitroperm VP800R stripes from Vacuumschmelze GmbH.

In this present paper, a high operation temperature (up to 220 °C) ring-core fluxgate magnetic sensor is designed. The core material, as well as the structural and coil materials, are carefully selected and processed. The operation parameters are carefully chosen to improve temperature stability. Through a set of specially designed validation and calibration experiments performed at a series of temperatures, an in-depth study of the sensor’s temperature characteristics is carried out.

## 2. Fabrication

The fluxgate sensor consisted of a magnetic core, an excitation coil, an induction coil, and a set of structural components, including a shaft and a coil frame. The excitation coil was wound on the magnetic core with a total of 54 turns, covering a complete circle of the core. The induction coil was wound on the coil frame, with a total of 414 turns in four layers. The combination of excitation coil and magnetic core was mounted to the coil frame via the shaft. These components formed an open-loop ring-core parallel fluxgate sensor, as shown in Figure 1.

The core material must be stable over a wide temperature range, and its Curie temperature should be well above the designed operating temperature. The model of the core was GMG1MN 151005. It was made by Shenzhen GMcore Technology Co., LTD. Its Curie temperature was 570 °C. At room temperature, its initial relative permeability was high, while its coercivity was low [18]. The core was wound from a nanocrystalline thin ribbon of approximately 20 μm, with an outer diameter of 14.5 mm, an inner diameter of 10.0 mm, and a height of 5.0 mm.

The temperature resistance of the original package of the core was 120 °C. The package was removed, and the core was injection-molded by epoxy resin, which was 260 °C resistant. The product code of the epoxy resin was YH9621A/B, which represents a 2-component epoxy resin produced by BOQIAO Electronic Materials Co., Ltd., Guangzhou. After curing at room temperature for 24 h, the mold was released, the surface was finely polished, and the re-packaged core was further stabilized at 180 °C for 1 h. Wrapped in epoxy resin, the finished magnetic core had an outer diameter of 15.5 ± 0.1 mm, an inner diameter of 9.0 ± 0.1 mm, and a height of 6.3 ± 0.2 mm.

The temperature resistance of the coil is also critical. For the excitation coil, a modified polyesterimide amide-enameled copper wire with a diameter of 0.5 mm was used. For the induction coil, a self-adhesive modified polyesterimide amide-enameled copper wire with a diameter of 0.13 mm was used. The enameled copper wires were produced by ELEKTRISOLA Dr. Gerd Schildbach GmbH & Co KG.; the product code of the wires was KS22. The typical resistance of the 0.13 mm wire was 1.288 Ω/m@20 °C, and the typical resistance of the 0.50 mm wire was 0.087 Ω/m@20 °C. Both could withstand an 220 °C operating temperature.

The structural components were made of engineering plastic polyetheretherketone (PEEK); its long-term functional temperature was 260 °C. The components were machined by milling. Polyimide high temperature tape was used to improve surface flatness and elasticity and to protect the coating of the coils.

As a whole, the sensor was 22 mm in length, 24.8 mm in width, 14.6 mm in height, and approximately 15 g in weight.

## 3. Operation Principle

Being a Goubau-type fluxgate sensor, the excitation coil generated a toroidal AC magnetic field driven by a waveform generator and a power amplifier in series. Most of its magnetic flux was confined in the magnetic core, which decreased the core magnetic loss. The excitation field was well beyond the threshold, saturating the core deeply, which triggered the fluxgate effect. The induction coil then picked up the accordant signal, which contained both the information of the excitation drive field and the external field to be measured. An oscilloscope was used to measure the induction signal. As a whole, the sensor resembled a nonlinear magnetic amplifier or transformer. In order to measure and analyze the signals during its operation, a circuit, shown in Figure 2, was built.

The output of the waveform generator and the power amplifier was a 20Vpp AC sinusoidal signal. It drove the magnetic core to saturate periodically. When the core was saturated, the inductance of the excitation coil decreased sharply, causing its impedance to instantly drop by several orders, resulting in a pulse waveform similar to a short-circuit, as shown in the red curve in Figure 3.

In Figure 3, the magnitude of the ambient field is reflected in the height change of the induction pulse signal. Theoretically, the fluxgate effect generates a secondary induction signal, the magnitude of which should be proportional to the ambient field [1]; the dominant frequency of the secondary signal should be equal to twice the main frequency of the excitation signal. In the case of this paper, the excitation signal and induction signal are very similar to pulses. The spectrum of the induction signal actually consists of two parts. One is from the discrete spectral lines caused by the pulses of the excitation signal, which is an odd number of times the main frequency of the excitation signal. The other part comes from the pulse discrete spectral lines of the double frequency generated by the fluxgate effect, which is an even number of times the main frequency of the excitation signal [1]. That is, each evenly-timed spectral line component dominantly corresponds to an oddly-time spectral line through the fluxgate effect. The operation and sampling modes are similar to the principle raised by Primdahl [19]. All even harmonic spectra deflect the flux density of the ambient magnetic field. A sampling sequence was generated to collect the peak phases of the induction signal exclusively [20]. This method of digital detection suppresses noise by discarding unnecessary signals [21].

Since the pulse signal will absorb a lot of power in a short period, a tuning capacitor was connected in parallel with the sensor to provide a high-peak current to the sensor [22], as shown by *C*_1_ in Figure 2.The excitation circuit should operate in an oscillating discharge state. The damping factor ξ can be derived as:
(1)$$\xi =\frac{R_{s}}{2}\sqrt{\frac{C_{1}}{L_{s}}}<1$$

To fulfill the damping factor restriction, the value of the tuning capacitor was:
(2)$$C_{1}<\frac{4\cdot L_{s(sat)}}{R_{s}}$$
where $L_{s(sat)}$ denotes the saturated inductance value of the sensor, and
$R_{s}$ denotes the series resistance of the sensor. At room temperature,
$L_{s(sat)}\approx 140\;\mu$H @ 1 kHz, $R_{s}\approx 3.2\;\Omega$. Therefore, *C*_1_ should be less than
55 μF. A variety of capacitor values from
1 μF to 10 μF were tested to determine the best-matching tuning capacitance for the sensor.

In Figure 4, the functional frequency value decreases with the capacitance increasing. The critical frequency is roughly and inversely proportional to the root of the capacitance, which is in line with the theoretical expectations. Note that the sharp valley shows where the meso-stable operating frequency for each capacitor is, and the instability points of 1.0 μF and 2.2 μF are not reflected in the figure. The resonant frequency of the RLC network is higher than the frequency shown. It can be found in Figure 4 that, for each capacitance, the higher the operating frequency, the closer to the resonant frequency, and the larger the signal intensity. As the frequency increases, the value of
$L_{s}$ gradually decreases, resulting in a damping factor
ξ that becomes gradually greater than 1. When the damping factor
ξ is approximately equal to 1, the aforementioned meso-stable state will occur. In the meso-stable state, due to the drive current source, the oscillating discharge state can be maintained but cannot be reconstructed. When the damping factor
ξ continues to increase with frequency, the drive current source cannot meet the power consumption, and the oscillating discharge state fails. Obviously, the
4.7 μF capacitor maximizes the intensity of the induced signal. Therefore, in the subsequent tests, the 4.7 μF capacitor was chosen as the tuning capacitor.

## 4. Testing and Analysis

To test the performance of the fluxgate sensor in the entire temperature range, a specially designed heating device was used to minimize the magnetic interference. The heating device consisted of an oil storage tank, a heating rod, an oil pump, an oil bath heating urn, a thermal insulation tank, and oil pipelines. The oil pipelines were made of high-temperature-resistant hose, 3 m long, connecting the heating urn and the oil storage tank where the heating rod and the pump were located. This kpt the sources of electromagnetic interference as far away as possible from the heating urn and the sample being tested inside it. The oil bath heating urn was a glass double-layered vessel through which the heating oil flowed. The thermal insulation tank was made of brass, filled with asbestos, and wrapped the heating urn. It was small enough to fit in a shielding cylinder or Helmholtz coils.

A series of tests on sensor impedance, hysteresis, permeability, and sensitivity characteristics were carried out at the temperatures regulated by the heating device above.

### 4.1. Impedance

Approximate to a nonlinear inductor, the impedance of the excitation coil can be equivalent to the series connection of an inductor and a resistor. The impedance *Z* and its phase angle
θ shows the frequency response of the sensor. In order to ensure the temperature stability, the change of
|Z| should be within an acceptable range. More importantly,
θ should be stable, so as to ensure that the sensor operates as an inductive element. The measured impedance and its phase are shown in Figure 5.

As shown in Figure 5, at different temperatures, the phase angles around 1 kHz are closest to
$90^{{}^{\circ}}$, meaning that the sensor is closest to an ideal inductive component. Therefore, the operation frequency of the sensor was set to 1kHz in the subsequent experiments.

### 4.2. Hysteresis Loops

The hysteresis loops are a reflection of the coercivity, saturated magnetic flux density, and the changes in the permeability of the magnetic material. They are beneficial for understanding the core’s magnetic-temperature properties to observe the relative changes caused by temperature.

Referring to the circuit in Figure 2, the strength of the magnetic field
$H_{E}$ generated by the excitation coil can be expressed as:
(3)$$H_{E}=n_{E}I_{E}=n_{E}\frac{U_{1}}{R_{1}}$$
where $n_{E}$ is the number of turns of the excitation coil, $I_{E}$ is the excitation current, $U_{1}$ is the voltage signal of VF1, and $R_{1}$ is the resistance of the sampling resistor R1 in Figure 2.

The electromotive force
$U_{S}$ on the induction coil originates from the change in the magnetic flux density $B_{S}$ of the magnetic core, assuming that:
(4)$$U_{S}=n_{S}\frac{dB_{S}\cdot A}{dt}$$
where $n_{S}$ is the number of turns of the induction coil, and *A* is the area of the coil. It should be noted that in the hysterisis test only, the induction coil is different from the one described above. It had 10 turns and was directly attached to the core along with the excitation coil, covering less than a quarter of the whole ring. Therefore, $B_{S}$ can be derived as:
(5)$$B_{S}=\frac{1}{n_{S}\cdot A}\int_{0}^{t}U_{S}dt+B_{0}$$
where $B_{0}$ is the initial value of the integral. Considering that the entire test was carried out in a shielding cylinder with a small magnetic field strength, and the operation frequency was relatively high, the influence of
$B_{0}$b was relatively small. The hysteresis loop can be represented as a Lissajous graph of $H_{E}$ and $B_{S}$.

As Figure 6 shows, the saturation flux density of the core decreases with increasing temperature. From room temperature to 230 °C, the saturation flux density decreases monotonically from approximately 0.44 T to 0.36 T. Despite this, the shape of the loops remains consistent, which proves that the fluxgate effect behaves similarly across the temperature range.

It should be pointed out that the flux density described above refers to the flux density within the cross-sectional area of the coil, not the flux density within the cross-sectional area of the core. Since the core is epoxy-potted, the area of the coils used for the hysteresis loop measurement, which was 40 mm${}^{2}$, is approximately 1.6 times the area of the core, which was 25 mm${}^{2}$. The magnetic flux density within the core should be the flux density within the coil multiplied by the area calibration factor $C_{area}=1.6^{2}=2.56$. The calibrated flux density is close to the value given in the manual of the core, which is 1.2 T. Here, for ease of understanding and calculating, the magnetic flux density within the cross-sectional area of the coil continues to be used.

The hysteresis loops shown in Figure 6 were all measured at deep-saturation excitation, which lacks detail near a zero magnetic field. Figure 7 gives a detailed hysteresis loop measured when the core is just saturated at 220 °C.

Figure 7 proves that, at 220 °C, the soft magnetic properties of the core remain good. The hysteresis loop morphology still meets the need of fluxgate applications.

### 4.3. Initial Magnetic Permeability

The magnetic permeability and its changing law are the key issues in the study of fluxgate sensors. Permeability determines the impedance of the sensor, which in turn determines the frequency characteristics of the circuit. The differentiation of the permeability is one of the coefficients of sensitivity [1].

For soft magnetic materials, permeability varies with the strength of the magnetic field [10]. The initial permeability $\mu_{i}$ refers to the permeability of the material when H=0. Correspondingly, the saturation permeability $\mu_{s}$ refers to the permeability when the material is saturated. $\mu_{s}$ is theoretically approximately equal to the air permeability, which is interfered by many environmental factors.

Practically, the initial permeability of the core is calculated indirectly by measuring its inductance with a small oscillating signal using an impedance analyzer. An Agilent Technologies E4990A Impedance Analyzer was used to measure the inductance of the excitation coil at different temperatures. The inductance *L* measured at a small signal can be expressed as: (6)$$L=\mu_{i}\mu_{0}\cdot \frac{A_{e}\cdot N^{2}}{L_{e}}$$
where $\mu_{0}$ refers to vacuum permeability, $A_{e}$ refers to the cross-sectional area of the core, *N* refers to the number of turns of the coil, and $L_{e}$ refers to equivalent magnetic path length; then: (7)$$\mu_{i}=\frac{L_{e}}{\mu_{0}A_{e}\cdot N^{2}}\cdot L$$

For the core GMG1MN 1510005, the data sheet indicates $A_{e}=8.8$ mm${}^{2}$ and $L_{e}=38.5$ mm. Substituting in these values and carrying out the measurement, we obtain the initial permeability of the core at 5 temperatures shown in Figure 8.

Unlike $B_{S}$, the initial permeability dose not vary monotonically with temperature in Figure 8. It gains a maximum around 75 °C, then slowly decreases by an order of magnitude. Nevertheless, it is still orders of magnitude larger than the air permeability at saturation.

Assuming that the core permeabilities $\mu_{i}$ and $\mu_{S}$ are both constants, then the saturation field strength $H_{S}$ can be expressed as: (8)$$H_{S}={\left.\frac{B_{S}}{\mu_{i}\mu_{0}}\right|}_{f=1kHz}$$

Refering to Equation (Equation 3), $I_{E}{|}_{saturate}$ is:
(9)$$I_{E}{|}_{saturate}=\pm {\left.\frac{B_{S}}{\mu_{i}\mu_{0}n_{E}}\right|}_{f=1kHz}$$
where $B_{S}$ can be obtained from Figure 6. The results are shown in the following Table 1.

It is clear that, since the required excitation current for the core to saturate differs, the exact phase at which the core goes into saturation is slightly different. This requires more attention for the sampling circuit when the sensor is put into practice. To improve high-temperature reliability, the sampling phase needs to track the saturation point on the excitation signal in real time rather than the common method of maintaining a fixed phase shift with the excitation signal source.

### 4.4. Calibration and Sensitivity

The calibration experiment was a two-step process. First, the sensor was calibrated by a solenoid in a seven-layered shielding cylinder at room temperature. Second, the sensor was calibrated by a Helmholtz coil in the oil bath heating urn to obtain the temperature-sensitivity relationship. The second step was carried out at the Geomagnetic Observatory in Beijing [23], where the magnetic field is calm. The first step was mainly to provide a benchmark reference for the second step.

In the shielding cylinder calibration, the directions of the sensor and the solenoid were aligned. With the 7-layered shielding cylinder sealed, a high-precision current source was used to control the solenoid to generate a DC magnetic field from $-1\times 10^{5}$ nT to $+1\times 10^{5}$ nT with a step of 5000 nT.

As shown in Figure 9, the output voltage is well proportional to the applied magnetic field. The linear fitting result is: (10)$${\hat{U}}_{S}=2.36376\times 10^{-5}\cdot B_{A}-6.58762\times 10^{-3}$$
where the unit of ${\hat{U}}_{S}$ is volts, and the unit of $B_{A}$ is nanoteslas. This gives the sensitivity of 23.6 kV/T@25 °C and an offset of 278.7 nT.

In the oil bath heating urn, the sensor was calibrated at different temperatures. The temperature points varied from 28 °C to 220 °C with a step of 10 °C. Figure 10 compares the results for 5 out of the 19 temperatures.

Due to the complex temperature test conditions, the probe could not be well aligned with the coil axis. The results shown in Figure 10 needed to be adjusted with reference to the linear fitting parameters obtained previously, including scale and offset. The linear fitting parameters adjusted by Equation (Equation 10) in Figure 10 are shown in Table 2.

A detailed table of the coefficients of linear fit covering all 19 temperatures is shown in Appendix A Table A1. By linearly fitting the sensitivity values of all 19 temperature points, it is concluded that the temperature drift of the sensor is 742 ppm$\cdot^{{}^{\circ}}$C${}^{-1}$ over the entire range.

## 5. Conclusions

A ring-core fluxgate magnetic sensor operating at temperatures up to 220 °C is designed. Aiming at high-temperature magnetic field detection, a type of Fe-based nanocrystalline that has high temperature resistance is used as magnetic core. High-operation-temperature-enameled wires, plastics, epoxy resin and tape are used to build coils and structural components. In-depth tests show that the sensor operates steadily from room temperature to 220 °C and even higher. The sensor works with a sensitivity of approximately 24 kV/T at 25 °C with an acceptable temperature coeficient of 742 ppm/°C throughout the range.

There are currently few public institutions that are able to provide a non-magnetic constant temperature environment of up to 220 °C. Though a set of specially designed heating devices is used to avoid magnetic interference, the magnetic environment for any of the tests in this paper involving heating is still complex. The pump and heating rod are sources of interference, and the heat of the pipeline would cause thermal deformation to shielding cylinders, Helmholtz coils, etc.

The operation frequency is 1kHz, which is due to the high inductance of the sensor. Replacing smaller cores of the same Fe-based nanocrystalline may increase the frequency, which is beneficial for improving the sampling bandwidth while further reducing the size of the sensor.

The potential to operate at higher temperatures [17] is limited by the nominal maximum operating temperature of the wires and epoxy resin. Materials with higher temperature resistance and better processes are under study. Noise, thermal drift, long-term drift and many other important parameters have not been tested due to the lack of a long-term observation system for this sensor. The advantages of the ring-core geometry compared with other geometries in terms of the signal-to-noise ratio [24] cannot be verified. This will be the focus of our next research.

## Figures and Tables

**Figure 1 micromachines-13-02158-f001:**
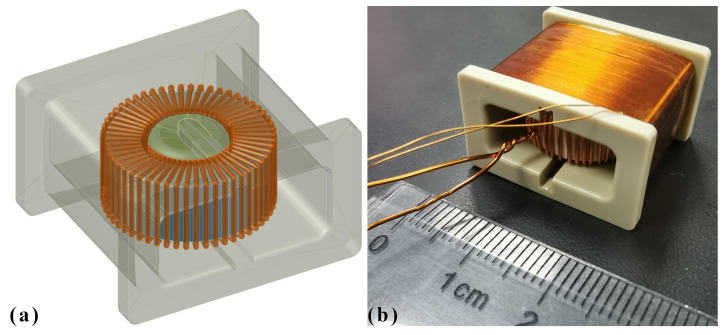
(**a**) A CAD view of the sensor’s core (blue), shaft (green), excitation coil (bronze), coil frame (grey), and how they are assembled. (**b**) A photo of the assembled sensor, with the induction coil wound on the frame and covered with high-temperature polyimide tape.

**Figure 2 micromachines-13-02158-f002:**
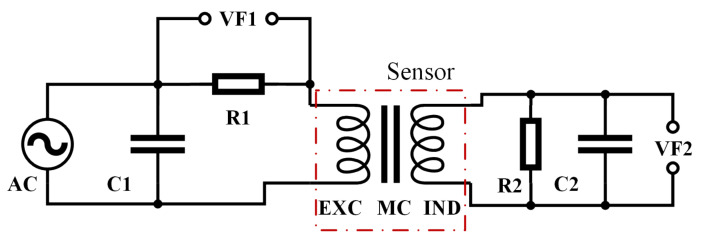
The test circuit of the sensor. AC represents the waveform generator and the power amplifier. C1 is the tuning capacitor. R1 is the excitation current sampling resistor. EXC is the excitation coil, MC is the magnetic core, IND is the induction coil, and the 3 components inside the dot-dashed red box as a whole represent the sensor. R2 and C2 are the loads of the induction coil. VF1 and VF2 are measuring points for oscilloscope.

**Figure 3 micromachines-13-02158-f003:**
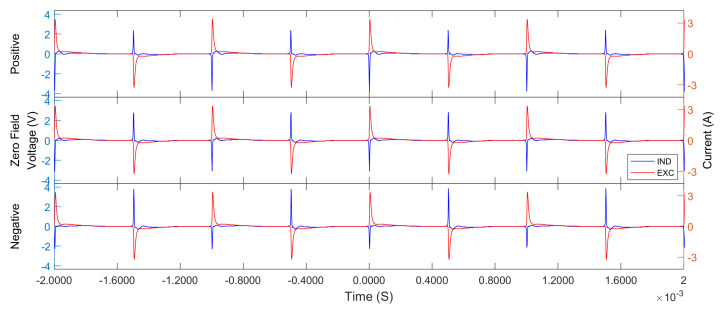
A sample of excitation signal versus induction signal in different ambient magnetic fields. The red curve named EXC represents the excitation current signal, and the blue curve named IND represents the induction voltage signal. The field was generated by Helmholtz coils in a shielding cylinder, which were equal in magnitude but opposite in direction. Zero-field condition is also shown.

**Figure 4 micromachines-13-02158-f004:**
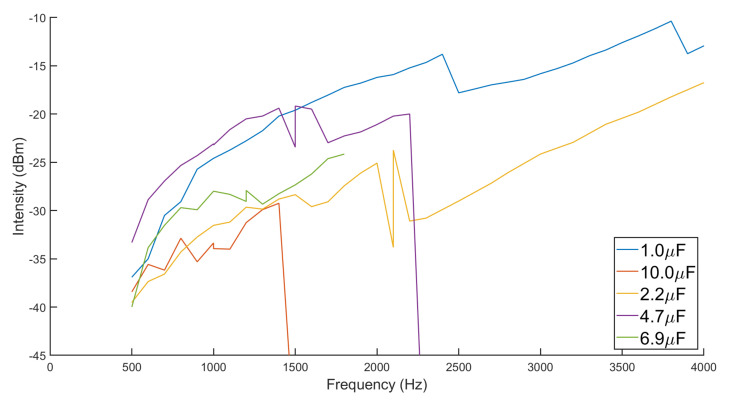
Induction signal amplitude under the same magnetic field versus different excitation frequencies with different tuning capacitances. The intensity power value represents the component of twice the excitation frequency in the induced signal. The intensity of the specific frequency is measured by an Agilent Technologies N9020A MXA Signal Analyzer.

**Figure 5 micromachines-13-02158-f005:**
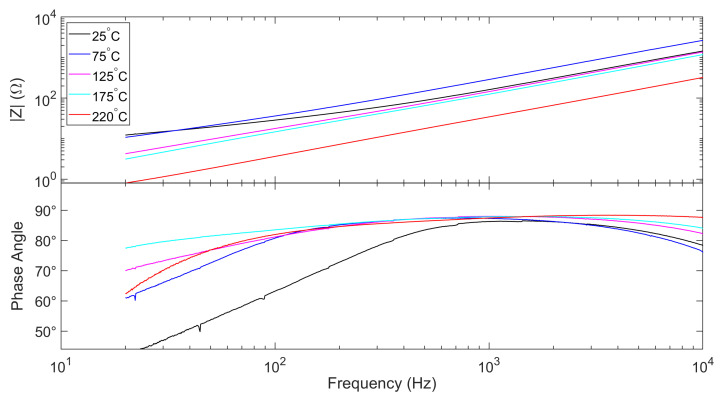
Impedance and its phase of the excitation coil measured under different temperatures. Different color indicates different sensor temperature. The values of |Z| at 1 kHz are 161.4 Ω at 25 °C, 286.74 Ω at 75 °C, 144.3 Ω at 125 °C, 124.5 Ω at 175 °C and 34.3 Ω at 220 °C.

**Figure 6 micromachines-13-02158-f006:**
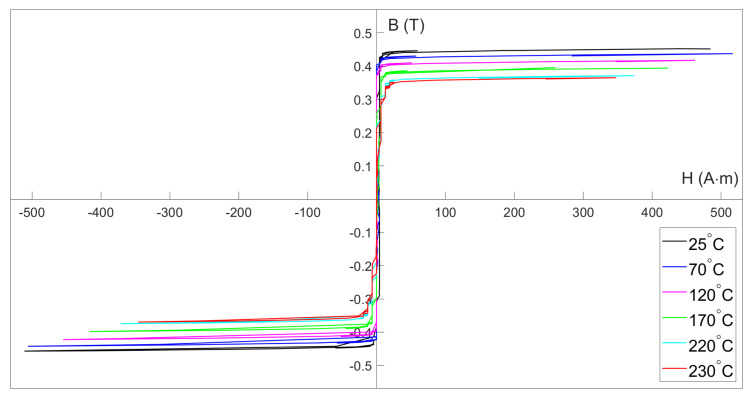
Hysteresis loop of the magnetic core measured at different temperatures at 1kHz. Different colors indicate different sensor temperature.

**Figure 7 micromachines-13-02158-f007:**
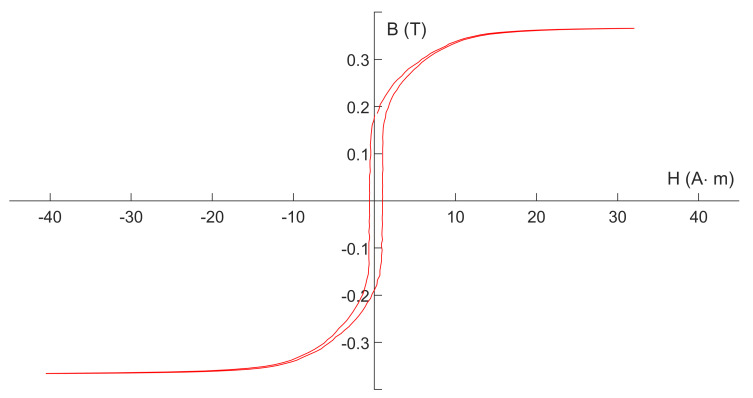
Hysteresis loop of the magnetic core measured at 220 °C.

**Figure 8 micromachines-13-02158-f008:**
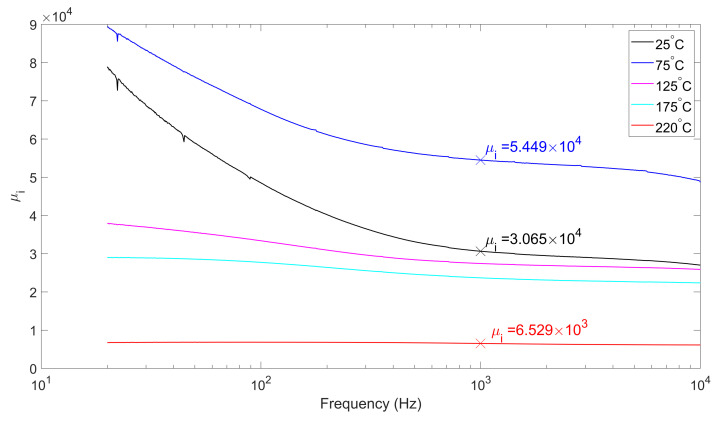
Initial permeability at different temperatures. The permeability at 1kHz is marked by cross.

**Figure 9 micromachines-13-02158-f009:**
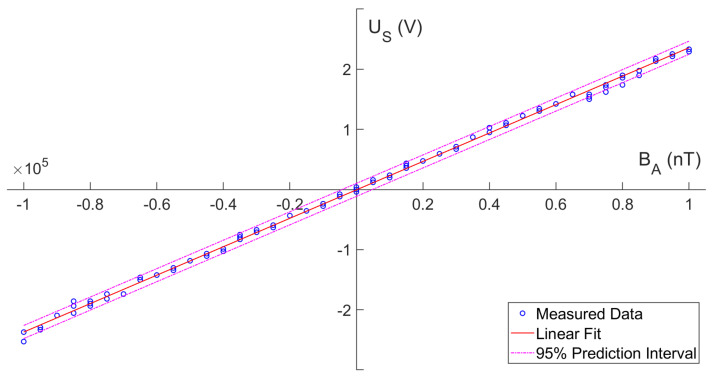
The output voltage $U_{S}$ versus the applied DC magnetic field $B_{A}$. The linear fitting curve is shown in red line. The 95% prediction interval is shown in magenta dotted lines.

**Figure 10 micromachines-13-02158-f010:**
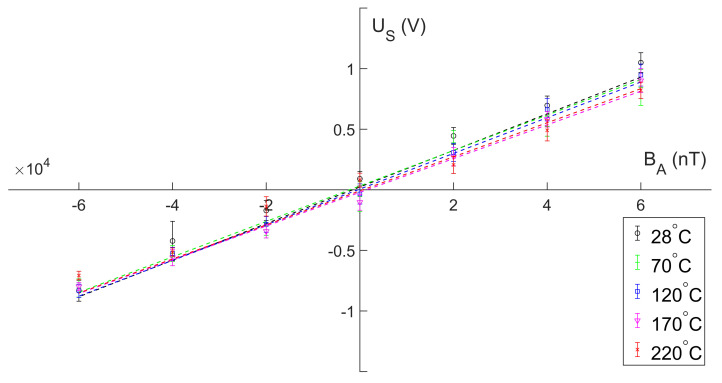
The output voltage $U_{S}$ versus the applied DC magnetic field $B_{A}$ at five of the temperature points. Data at different temperatures use different scatter markers and colors to be distinguished.

**Table 1 micromachines-13-02158-t001:** When the core saturates at different temperatures, the approximate value of the magnetic flux density, and the corresponding field strength and excitation current.

Temperature	$B_{S}\left(T\right)$	$\mu_{i}$	$H_{S}\left(A\cdot m\right)$	$I_{E}{|}_{\mathrm{saturate}}\left(A\right)$
25 °C	0.440	30,648	11.425	0.2116
75 °C	0.429	54,490	6.265	0.1160
125 °C	0.409	27,452	11.856	0.2196
175 °C	0.380	23,685	12.767	0.2364
220 °C	0.360	6529	43.878	0.8126

**Table 2 micromachines-13-02158-t002:** Normalized parameters of temperature calibration test. The parameters in the table correspond to ${\hat{U}}_{S}=C_{1}\cdot B_{A}-C_{0}$.

Temperature	28 °C	70 °C	120 °C	170 °C	220 °C
$C_{1}\left(\times 10^{-5}\left(V/nT\right)\right)$	2.36376	2.17768	2.25592	2.18938	1.96660
$C_{0}\left(\times 10^{-3}\left(V\right)\right)$	6.58762	50.33063	36.00119	83.97571	60.79530

## Data Availability

Publicly available datasets were analyzed in this study. This data can be found here: https://github.com/Yuankx13/220HightempDataSet.

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
