# Peer review of "Ring-Core Fluxgate Sensor for High Operation Temperatures up to 220 °C"

_micromachines, 2022, doi:10.3390/mi13122158_

Round 1

Reviewer 1 Report

The authors show a ring-core type fluxgate sensor with a high temperature consistency up to 220 °C, which is achieved by using Fe-based nanocrystalline magnetic core, 4 PEEK structural components, a epoxy resin wraping as well as a broad band short-circuited working 5 mode. The reviewer suggests to publish the manuscript after minor revisions.

1. In the line 46, coli should be coil.

2. Language should be checked carefully, such as in the line 109, the part of roughly inversely should add and between the two words.

3. In the introduction, the authors could give more details how the materials and structures are selected, such as the difference between Ferro-based nanocrystalline and other core materials etc.

Author Response

Dear reviewer,

Thank you for reviewing my manuscript!  Here is my reply to the review report.

  1. Line 46, the spell of 'coil' is corrected.
  2. The word 'and' has been added.
  3. The following description have been added to line 36 to clarify the main differences between Ferro-based nanocrystalline and other materials:

'Some models of Fe-based nanocrystalline materials have a Curie temperature of nearly 600℃. Their crystal stability is also better than that of amorphous alloys, and the crystallization temperature exceeds 500℃. More importantly, their good soft magnetic properties made them suitable core materials for fluxgates.'

And a corresponding reference has been added. In short, the main advantages of Fe-based nanocrystalline originates from its micro and macro structure. It has short-range orderliness and forms nanoscale crystallization. At the macroscopic scale, the orientation of these crystals is random and leads to highly isotropic. On this basis, it obtains good soft magnetic properties.

Thank you again for your comments!

Reviewer 2 Report

The manuscript introduces a new high-temperature resistant prototype of fluxgate magnetometer. It can withstand up to 220 degrees Celsius, which is one of the highest reported operating temperatures for fluxgates. Such fluxgates can take on the task of monitoring and analyzing magnetic fields in deep directional drilling at depths of more than 5,000 meters. A commercial-grade nanocrystalline soft magnetic material is applied as the core, and a series of supporting processing techniques are used to improve the overall high temperature resistance. More importantly, the prototype is characterized through out the whole temperature range. The evaluation results reflect several key properties to guide not only the optimization of this specific prototype, but also to guide other fluxgate designs if they are required to improve temperature reliability.

Before acceptance, the following questions should be answered, and a few contents should be clarified.

1.     In Figure. 4, There are multiple glitches and collapses. These signs clearly have special implications, but they are described very briefly. The changing law of the induction signal intensity versus different tuning capacitors and different working frequencies is a key character to judge whether the resonance net is functional. So, the meaning of the curves in figure 4 should be more detailed, and the relationship between damping factor and the resonance net should be clarified.

2.     In Figure. 6, The cluster of hysteresis loops at different temperatures is instructive. However, it is quite blurry near zero magnetic field. Giving the initial information about the hysteresis loop, including initial permeability and coercivity, is equally important as showing permeability changes and saturation state. Therefore, it will be better to show a local hysteresis diagram at near zero field.

3.     In Table. 1, The saturated flux intensity shown is around 0.440T. There is obviously a gap comparing to the values given in the manual. What causes this gap? Can this gap be corrected?

4.     In the manuscript, there are some spelling mistakes as listed below. Please check carefully.

1)       Line 40, ‘carfully’ should be ‘carefully’.

2)       Line 46, ‘an excitation coli’ should be ‘an excitation coil’.

3)       Line 70, ‘structrural’ should be ‘structural’.

4)       Line 72, as far as I know, ‘Polymide’ should be ‘Polyimide’.

5)       Line 77, ‘a totoidal AC magnetic field’ should be ‘a toroidal AC magnetic field’.

6)       Line 108, ‘increaseing’ should be ‘increasing’.

7)       Figure. 5, the figure needs to be re-shaped, because the x-label is not complete.

8)       Line 183, ‘stll’ should be ‘still’.

9)       Line 225, ‘coeficient’ should be ‘coefficient’.

10)   Line 230, ‘sourses’ should be ‘sources’.

Author Response

Thank you for your review. Please see the attachment for the reply of the comments.

Author Response

Thank you for your comments. Please see the attachment for a point-to-point response!
